# Neonatal Birthweight Spectrum: Maternal Risk Factors and Pregnancy Outcomes in Saudi Arabia

**DOI:** 10.3390/medicina60020193

**Published:** 2024-01-23

**Authors:** Hayfaa Wahabi, Hala Elmorshedy, Yasser S. Amer, Elshazaly Saeed, Abdul Razak, Ibrahim Abdelaziz Hamama, Adnan Hadid, Samia Ahmed, Sarah A. Aleban, Reema Abdullah Aldawish, Lara Sabri Alyahiwi, Haya Abdullah Alnafisah, Raghad E. AlSubki, Norah Khalid Albahli, Aljohara Ayed Almutairi, Layan Fahad Alsanad, Amel Fayed

**Affiliations:** 1Research Chair for Evidence-Based Health Care and Knowledge Translation, King Saud University, Riyadh 11451, Saudi Arabia; hwahabi@ksu.edu.sa (H.W.); yassersamiamer@gmail.com (Y.S.A.); sesmaeil@ksu.edu.sa (S.A.); 2Department of Family and Community Medicine, College of Medicine, King Saud University, Riyadh 11451, Saudi Arabia; 3Department of Tropical Health, High Institute of Public Health, Alexandria University, Alexandria 5424041, Egypt; hiph.halanasser@alexu.edu.eg; 4Clinical Practice Guidelines and Quality Research Unit, Corporate Quality Management Department, King Saud University Medical City, Riyadh 11451, Saudi Arabia; 5Prince Abdulla Bin Khaled Coeliac Disease Research Chair, Department of Pediatrics, College of Medicine, King Saud University, Riyadh 11451, Saudi Arabia; ehamed@ksu.edu.sa; 6Neonatal Intensive Care Unit, Department of Pediatrics, Princess Nourah Bint Abdulrahman University, Riyadh 11564, Saudi Arabia; abdul.razak@monash.edu (A.R.); iahamama@kaauh.edu.sa (I.A.H.); 7Neonatal Intensive Care Unit, Department of Pediatrics, King Saud University Medical City, Riyadh 11451, Saudi Arabia; aalhadid@ksu.edu.sa; 8Clinical Sciences Department, College of Medicine, Princess Nourah Bint Abdulrahman University, Riyadh 11564, Saudi Arabia; saraalaiban99@gmail.com (S.A.A.); aldawishreema@gmail.com (R.A.A.); alyahiwilara.s@hotmail.com (L.S.A.); hayaabdullah939@gmail.com (H.A.A.); raalsubki@gmail.com (R.E.A.); norah.k.b@hotmail.com (N.K.A.); aljoharaayedhm@gmail.com (A.A.A.); 9College of Medicine, King Saud University, Riyadh 11451, Saudi Arabia

**Keywords:** birth weight, maternal hypertension, maternal obesity, maternal diabetes, preterm birth, neonatal respiratory distress, Saudi Arabia

## Abstract

***Background and Objectives:*** Low-birth-weight (LBW) neonates are at increased risk of morbidity and mortality which are inversely proportional to birth weight, while macrosomic babies are at risk of birth injuries and other related complications. Many maternal risk factors were associated with the extremes of birthweight. The objectives of this study are to investigate maternal risk factors for low and high birthweight and to report on the neonatal complications associated with abnormal birth weights. ***Materials and Methods:*** We conducted a retrospective analysis of medical records of deliveries ≥ 23 weeks. We classified the included participants according to birth weight into normal birth weight (NBW), LBW, very LBW (VLBW), and macrosomia. The following maternal risk factors were included, mother’s age, parity, maternal body mass index (BMI), maternal diabetes, and hypertension. The neonatal outcomes were APGAR scores < 7, admission to neonatal intensive care unit (NICU), respiratory distress (RD), and hyperbilirubinemia. Data were analyzed using SAS Studio, multivariable logistic regression analyses were used to investigate the independent effect of maternal risk factors on birthweight categories and results were reported as an adjusted odds ratio (aOR) and 95% Confidence Interval (CI). ***Results:*** A total of 1855 were included in the study. There were 1638 neonates (88.3%) with NBW, 153 (8.2%) with LBW, 27 (1.5%) with VLBW, and 37 (2.0%) with macrosomia. LBW was associated with maternal hypertension (aOR = 3.5, 95% CI = 1.62–7.63), while increasing gestational age was less likely associated with LBW (aOR = 0.51, 95% CI = 0.46–0.57). Macrosomia was associated with maternal diabetes (aOR = 3.75, 95% CI = 1.67–8.41), in addition to maternal obesity (aOR = 3.18, 95% CI = 1.24–8.14). The odds of VLBW were reduced significantly with increasing gestational age (aOR = 0.41, 95% CI = 0.32–0.53). In total, 81.5% of VLBW neonates were admitted to the NICU, compared to 47.7% of LBW and 21.6% of those with macrosomia. RD was diagnosed in 59.3% of VLBW neonates, in 23% of LBW, in 2.7% of macrosomic and in 3% of normal-weight neonates. Hyperbilirubinemia was reported in 37.04%, 34.21%, 22.26%, and 18.92% of VLBW, LBW, NBW, and macrosomic newborns, respectively. ***Conclusions:*** Most neonates in this study had normal birthweights. Maternal hypertension and lower gestational age were associated with increased risk of LBW. Additionally, maternal obesity and diabetes increased the risk of macrosomia. Neonatal complications were predominantly concentrated in the LBW and VLBW, with a rising gradient as birthweight decreased. The main complications included respiratory distress and NICU admissions.

## 1. Introduction

Birth weight is one of the most important determinants of neonatal survival and morbidity [1,2,3]. Globally, more than 35 million live births are classified as low birth weight (LBW) [4]. The prevalence of LBW is far higher in low- and middle-income countries compared to high-income countries [5]. There is significant heterogeneity in the clinical studies with regard to the prevalence of LBW in Saudi Arabia [1,6,7]. However, analysis from Riyadh Mother and Baby Multicenter Cohort study (RAHMA), reported a prevalence of 9.6% for birth weights less than 2.5 kg and 1.7% for birthweights less than 1.5 kg (very low birth weight (VLBW)) [8].

Macrosomia, defined as a birthweight ≥ 4.0 kg is found in 3.1% of the total deliveries in Saudi Arabia [8]. This is far less than the prevalence of 10–12% reported in other populations [9]. However, it is consistent with the wide range of instances reported by the recently published systematic review [10].

Several risk factors have been associated with LBW and VLBW including short gestation, multiple gestations, low maternal body mass index (BMI), and chronic maternal conditions such as hypertension and diabetes [7]. However, the most significant association is between VLBW and premature birth. Similarly, there is an association between macrosomia and post-term deliveries [11,12]. Compared to normoglycemic mothers, mothers with diabetes have a two to fivefold increased risk of having a macrosomic baby [13]. Other risk factors include previous history of fetal macrosomia, maternal obesity, and excessive gestational weight gain.

Compared to normal-weight neonates, those with LBW are at increased risk of morbidity and mortality which are inversely proportional to the birth weight [11]. However, there was a significant improvement in the survival rate of VLBW neonates over the last two decade from 67% to 90%. The improvement in survival rates was observed particularly in neonates weighing 750 to 1000 g, with a gestational age between 23 and 27 weeks [14]. However, despite the improvement in neonatal survival of LBW and VLBW, there is only modest improvement in neurodevelopmental complications rates [15]. A recently published report from Saudi Arabia showed that 39.2% of VLBW neonates were developmentally delayed, 36.2% had cerebral palsy, and 33.3% had speech problems [16].

On the other hand, macrosomic babies are more likely to experience birth injuries, shoulder dystocia, and an increased risk of obesity and metabolic disorders later in life. Mothers of a macrosomic baby are at risk of cesarean section delivery, postpartum hemorrhage, and birth canal injuries [10].

The objectives of this study are to investigate maternal risk factors for low and high birth weight and to report on the neonatal complications associated with the abnormal birth weight.

### Study Population

We conducted a retrospective analysis of a cohort of deliveries that occurred from 1 January 2021 to 31 December 2022 at King Abdullah bin Abdulaziz University Hospital (KAAUH), Princess Nourah bint Abdulrahman University (PNU). The hospital has 406 beds, excellent centers in women’s health, pediatric, and adolescent health, as well as a child-development center specialized in autism and attention disorders.

We included all single pregnancies with gestational age ≥ 23 weeks based on last menstrual period and/or ultrasound scanning. Twin pregnancies were excluded from this study and cases with missing birth weight measurements were also excluded. A total of 1855 cases met the eligibility criteria for the study. We classified all included cases according to birth weight into normal birth weight, LBW, VLBW, and macrosomia.

## 2. Materials and Methods

The following variables were collected: maternal risk factors; including the mother’s age, number of deliveries (parity), BMI at the time of delivery, presence of co-morbidities including pre-gestational diabetes mellitus (PGDM), gestational diabetes mellitus (GDM), hypertensive disorders of pregnancy, and attendance of antenatal care.

The neonatal outcomes included birthweight, APGAR score at 5 min, neonatal intensive care unit admission (NICU), diagnosis of respiratory distress (RDS), diagnosis of transient tachypnea of the newborn (TTN), the need for immediate resuscitation, the need for mechanical ventilation, umbilical cord arterial acidosis, initiation of breast feeding, feeding difficulty, need for continuous positive airway pressure (CPAP) or high-flow nasal cannula (HFNC) for more than 2 h, the need for FiO_2_ > 30% for 4 h or more, surfactant administration, neonatal hyperbilirubinemia, sepsis, and death.

### 2.1. Definitions

#### 2.1.1. Neonatal

**Normal birth weight (NBW)** is defined a birth weight between 2500 and 3999 g [17].**Low birth weight (LBW)** is defined as a birth weight of less than 2500 g.**Very low birth weight (VLBW)** is defined as a birth weight below 1500 g [17].**Macrosomia** is defined as a birth weight ≥ 4 kg [17].**Respiratory distress syndrome (RDS)** “Newborn babies with RDS have: (1) An arterial oxygen tension (PaO2) < 50 mm Hg and central cyanosis in room air, a requirement for supplemental oxygen to maintain PaO2 > 50 mm Hg, or a requirement for supplemental oxygen to maintain a pulse oximeter saturation > 85%. (2) A characteristic chest radiographic appearance (uniform reticulogranular pattern to lung fields and air bronchogram) within the first 24 h of life. The clinical course of the disease has been changed because of advances in treatment practices, including the use of early Continuous Positive Airway Pressure (CPAP)” [18].**Transient tachypnea of the newborn (TTN)** is a benign, self-limited RDS of term and late-preterm neonates related to delayed clearance of lung liquid. The distress appears shortly after birth and usually resolves within 3 to 5 days [18].**Neonatal hyperbilirubinemia (or neonatal jaundice)** results from elevated total serum bilirubin (TSB) and clinically manifests as yellowish discoloration of the skin, sclera, and mucous membrane [19].**Gestational age** at birth is defined as the time span between the conception and birth of an infant, calculated from the last menstrual period and/or early ultrasound scan when there is a difference between menstrual date and ultrasound date; the latter was taken as the correct date. It is further classified as follows [20]:Term pregnancy 39–41 weeksPost-term ≥ 42 weeksEarly-term pregnancy 37–38 weeksLate preterm 34–36 weeksEarly preterm < 34 weeks—it includes early preterm (32–33 weeks), very early preterm (28–31 weeks), and extremely early preterm (<28 weeks).Stillbirth: non-living birth ≥ 23 weeks of gestation [21].

#### 2.1.2. Maternal

**Maternal BMI** was calculated from maternal weight at delivery and height with the following cutoff values as suggested by Catalano et al. [22]: normal BMI (≤28.4 kg/m^2^), overweight (28.5–32.9 kg/m^2^), and obese (≥33 kg/m^2^).**Gestational diabetes mellitus (GDM)** is diagnosed at any time in pregnancy according to World Health Organization guidelines if one or more of the following criteria are met [23]:Fasting plasma glucose 5.1–6.9 mmol/L (92–125 mg/dL).1 h plasma glucose ≥ 10.0 mmol/L (180 mg/dL) following a 75 g oral glucose load.2 h plasma glucose 8.5–11.0 mmol/L (153–199 mg/dL) following a 75 g oral glucose load.**Pre-gestational diabetes mellitus (PGDM)** is a condition in which the mother has diabetes (most commonly type 1 or type 2 diabetes) before the onset of pregnancy [24].**Hypertensive events during pregnancy** according to the report of the national high blood pressure [25]: Pre-eclampsia is defined as new onset of elevated blood pressure after 20 weeks of pregnancy in a previously normotensive woman (≥140 mm Hg systolic or ≥90 mm Hg diastolic on at least two occasions 6 h apart) in addition to proteinuria of at least 1+ on a urine dipstick or ≥300 mg in a 24 h urine collection. Eclampsia is defined as seizures in a pre-eclamptic woman that cannot be attributed to other causes. Gestational hypertension is defined as a new onset of elevated blood pressure (≥140 mm Hg systolic or ≥90 mm Hg diastolic on at least two occasions 6 h apart) after 20 weeks of gestation in a previously normotensive woman, and defines superimposed pre-eclampsia as new onset of pre-eclampsia after 20 weeks of pregnancy. For this study, and due to the low prevalence of these conditions, all cases who developed gestational hypertension, pre-eclampsia, or eclampsia were aggregated in one category

### 2.2. Statistical Analysis

Data were analyzed using SAS Studio (SAS Institute Inc. 2015. SAS/IML^®^ 14.1 User’s Guide. Cary, NC, USA: SAS Institute Inc.), Categorical variables were summarized as frequencies and percentages and numerical variables were summarized as means ± standard deviation (SD). Statistical significance was tested by Chi square test, Fisher’s exact test, ANOVA test as appropriate.

Multivariable logistic regression analyses were carried out to investigate the independent effect of maternal factors on birthweight categories. We included the following risk variables: gestational age, mother age, maternal BMI, parity, hypertensive disorders during pregnancy, and maternal diabetes including both GDM, PGDM. The reference group was babies with normal birthweights (2500−<4000 g). The adjusted odds ratios (aOR) and 95% confidence intervals (95% CI) of the models were reported, along with area under the curve (AUC) of the Receiver operating characteristic curve (ROC) to reflect model performance. A tow-tailed test with *p* value < 0.05 was considered statistically significant.

### 2.3. Ethical Consideration

This study was approved by the Institutional Review Board of PNU, and followed the principles of Helsinki declaration. All data were anonymized and kept confidential. No identifiable information was included in the final manuscript. Written informed consent was waived for this study since retrospectively retrieved clinical data from medical records were used.

## 3. Results

We included 1855 cases in this study. There were 1638 neonates (88.3%) with NBW, 153 (8.2%) neonates with LBW, 27 (1.5%) cases with VLBW, and 37 (2.0%) neonates with macrosomia.

Bivariate analysis showed that diabetes and maternal obesity were associated with higher incidence of macrosomia as compared to neonates of mothers who were normoglycemic and with normal BMI (Table 1).

Gestational hypertension was reported more frequently among LBW (11.8%) and VLBW (19.2%) when compared to NBW (2.2%) and this trend was statistically significant (*p* < 0.01). While booking for antenatal care was significantly associated with lower incidence of LBW and VLBW, maternal age and parity did not show significant associations with low-birthweight categories or with macrosomia (Table 1).

More than 68% of NBW neonates were full term, and 26.6% of them were early term. The proportion of full term was substantially reduced among the LBW to 18.7% and to 3.7% in the VLBW neonates. Meanwhile, 13.3% of the LBW neonates were early term, and most of those with VLBW (88.9%) were classified as early preterm. In neonates with macrosomia, the majority were full term, 21.6% were early term, and 2.7% were post-term, as shown in Table 1.

Table 2 summarizes the adverse neonatal outcomes by birth-weight category. It is evident that adverse outcomes are clustered in neonates with LBW and VLBW. The results revealed that up to 81.5% of VLBW babies were admitted to the NICU, compared to 47.7% of LBW and 21.6% of newborns with macrosomia. Respiratory distress was reported more frequently among VLBW and LBW (59% and 23% respectively), while it was less reported among NBW and macrosomic infants (3% and 2.7% respectively). Similarly, immediate resuscitation and mechanical ventilation were most common in VLBW babies, followed by LBW and is least common in macrosomic babies. In addition, oxygen therapy as measured by CPAP/HFNC > 2 h and Baby FIO2 > 30% ≥ 4 h was notably higher in VLBW babies. Furthermore, VLBW newborns had the highest percentage of surfactant requirement. Tachypnea was most common in macrosomic neonates, followed by LBW and VLBW. Nine cases experienced arterial acidosis, of which eight cases were of normal birth weight, and one case was of VLBW (Table 2).

In total, 28 newborns had sepsis, with the highest percentage occurring in VLBW newborns. Hyperbilirubinemia was reported in 37.04%, 34.21%, 22.26%, and 18.92% of VLBW, LBW, NBW, and macrosomic newborns, respectively. CS was conducted in 29.3% in NBW, compared to 45.8% in LBW, and in more than 50% in VLBW, and nearly in 50% in Macrosomia. It is worth mentioning that there were seven stillbirths, four in VLBW, two in LBW, and one in NBW, as presented in Table 2. The mean APGAR score was considerably low (6.2 ± 1.2) in VLBW newborns, while it was comparable for the other birth weight categories. The mean gestational age was 36 ± 2 in LBW, compared to 29 ± 4 for VLBW and 40 ± 1 for newborns with macrosomia (Table 2).

Breast feeding was initiated in all groups with the highest proportion among NBW neonates (72.3%) and least among the VLBW (33.3%) (Table 2).

The summary of the characteristics of perinatal deaths is shown in Table 3. There were nine stillbirths and four neonatal deaths among the study population, which gives a perinatal mortality rate of 7.9/1000. Nine deaths (60%) were males, and the most frequent causes of perinatal mortality were preeclampsia, abruptio placenta, and congenital malformations. Only four (26%) of the deaths were ≤28 weeks, the rest were either at term (20%) or late preterm (53%). Nine of the mothers who had perinatal deaths were in their first pregnancy and the other four were of low parity (Table 3).

The regression analyses showed independent association between LBW and hypertensive disorders during pregnancy (OR = 3.5, 95% CI = 1.62–7.63); in contrast, increasing gestational age was less likely to be associated with LBW (OR = 0.51, 95% CI = 0.46–0.57), and the ROC curve for the LBW regression model had an AUC of 0.84 (95% CI = 0.80–0.88). Macrosomia was associated with maternal diabetes (OR = 3.75, 95% CI = 1.67–8.41), in addition to maternal obesity (OR = 3.18, 95% CI = 1.24–8.14), and the ROC curve of this model had an AUC of 0.75, 95% CI = 0.67–0.83. The odds of VLBW were reduced significantly with increasing gestational age (OR = 0.41, 95% CI = 0.32–0.53), and the ROC curve of the VLBW model had an AUC of 0.97, 95% CI = 0.96–1.00 (Table 4).

## 4. Discussion

The objectives of this study were to examine the association between maternal risk factors and abnormal birthweight, as well as the impact of birthweight on neonatal outcomes. Most of the neonates included in this study were of normal birthweight (88%).

The risk of LBW significantly increased with maternal hypertension and lower gestational age, while maternal obesity/overweightness reduced the occurrence of LBW (Table 4). Furthermore, we found increased risk of macrosomia with maternal obesity, and GDM/PGDM. Most of the neonatal complications were clustered in the categories of LBW/VLBW, with increasing gradient as birthweight decreases (Table 2). In addition, the results showed a perinatal mortality rate of 7.9/1000, and most of the deaths were due to obstetric causes.

The prevalence of the different birthweight categories and gestational age at delivery has not changed since it was last reported in 2016 [8,26] from Saudi Arabia in the RAHMA cohort study. Although these indices are similar to those reported in the gulf region [27] and in other high-income countries [28,29], they may indicate slow improvement in the perinatal care quality and uptake. It is worth noting that fewer mothers who delivered LBW neonates were booked for antenatal care in this study compared to mothers who had NBW neonates. Frequency and contents of antenatal care were proven to reduce the prevalence of LBW and its complications [30,31,32].

The prevalence of macrosomia is unexpectedly low considering the high prevalence of maternal obesity, GDM, and PGDM [8] among Saudi mothers and the high predictability of these risk factors for macrosomia as is evident from the area under the curve in Figure 1.

In addition, previous studies from Saudi Arabia showed significant association between maternal obesity and diabetes and macrosomia [33,34]. However, the low prevalence of macrosomia noted in this study and previous reports may be explained by the lower reference weight centiles of the Saudi Arabian neonates’ growth curve compared to curves from Europe and the USA [35,36] which makes maternal diabetes and obesity the main cause of fetal macrosomia with no other factors such as genetic factor or multiparity contributing to this category of birthweight [37,38].

Recently published studies showed that APGAR scores of less than seven at five minutes after delivery are significant predictors of cerebral palsy and neonatal mortality [39,40]. In this study, only neonates of VLBW had mean APGAR scores less than seven at five minutes (Table 2).

The effect of preterm birth (PTB) as a risk factor of LBW and VLBW is undeniable as shown in Table 4. PTB is the strong predictor of VLBW, with high sensitivity and specificity, as shown in the ROC model (Figure 1). Furthermore, and consistent with previous reports [41,42], our results showed that LBW is associated with maternal hypertensive disorders during pregnancy.

There is a proven inverse relationship between gestational age at delivery and neonatal survival and complications of prematurity and hence LBW [1,27]. Many of the neonatal complications we observed in this study (Table 2), were due to the immature respiratory system, liver, and immune system of the preterm neonates [43,44]. Hence, the higher proportion of neonates who suffer from respiratory distress, jaundice, and sepsis as shown in this study.

It is noticeable that the need for assisted ventilation and other therapeutic support is inversely proportional to the birthweight of the infant. However, our results showed that the highest proportion of TTN was observed among macrosomic neonates (Table 2). This finding agrees with previous publications that show TTN is common in macrosomic neonates delivered by cesarean section after prolonged labor and those born to diabetic mothers [45,46].

Most of the cases of acidosis at delivery in this study were observed in neonates of NBW, which is probably due to complications of labor and delivery.

The CS rate observed in this study for NBW neonates is comparable to that reported in previous studies from Saudi Arabia [8,47]. Nevertheless, we found that more than 50% of those with VLBW and nearly 50% of macrosomic neonates were delivered using CS (Table 2) [48,49]. The high proportion of CS delivery in LBW neonates can be explained partially by the complication of malpresentation which is common in preterm and in LBW [49] neonates. In addition, CS delivery has been supported by studies showing improved survival with less morbidity when compared to vaginal delivery [50]. However, such improvement in VLBW outcomes with the mode of delivery was not consistent and other studies showed no difference in outcomes based on the mode of delivery [48,51]. The CS delivery rate for macrosomic neonates was almost double that observed in NBW neonates, which is consistent with findings from previous studies both from Saudi Arabia [33] and other countries [52,53]. The main indications for CS delivery for macrosomic neonates reported in the literature are failure to progress in labor and previous CS delivery [54]. It is worth noting that vaginal delivery of a macrosomic infant is associated with higher risk of shoulder dystocia, infant birth trauma, maternal injuries, and postpartum hemorrhage, when compared to the same mode of delivery for NBW neonates [9].

While with more than 70% of NBW neonates breastfeeding was initiated, the proportions were significantly less in other categories of birthweight (Table 2). This can be explained by the need for admission to NICU and the mode of delivery by CS where mothers may need opiate analgesics and were not able to initiate breastfeeding as observed in other populations [55,56]. However, the association between NICU admission and failure to initiate breastfeeding was not consistent in the published literature [57,58].

The perinatal mortality rate reported in this study is notably less than that reported previously in Saudi Arabia less than a decade ago [8,59]. This drop in perinatal mortality maybe due to the effects of the pre-marital screening program and the associated reduction in congenital malformations in addition to improvement in antenatal and neonatal care. However, we cannot exclude a non-generalizable result as we collected the data from one hospital which is not a referral center and from which complicated cases detected during the antenatal period will be referred to specialized centers.

### Strength and Limitations

The study included large number of participants which makes the findings generalizable to other centers in Riyadh. It is one of the few studies in Saudi Arabia that provides important data on maternal risks associated with LBW and macrosomia and the outcomes associated with different birthweight categories. In addition, the study highlighted clear areas for further research such as the risk factors and outcomes of the sub-category of neonates who are small for their gestational age compared to those of LBW. In addition, further research is needed on the determinants of the initiation of breastfeeding in the Saudi community for all categories of birthweight and mode of delivery. Conducting longitudinal studies could also explore the developmental outcomes of neonates born with LBW, VLBW, and macrosomia, providing valuable insights into their long-term health and well-being.

However, it is important to acknowledge certain limitations of the study including the retrospective design which hinders the establishment of causal relationships between maternal risk factors and birth weight outcomes. Additionally, the study focuses on one region in Saudi Arabia which may restricts the generalizability of the findings to other regions. Moreover, due to the lack of available data, the study did not identify other potential risk factors such as maternal education, maternal smoking, interpregnancy interval, medications, and medical conditions other than diabetes and hypertension.

## 5. Conclusions

Most neonates in this study had a normal birthweight. Maternal hypertension and lower gestational age were associated with an increased risk of LBW, Additionally, maternal obesity and diabetes increased the risk of macrosomia. Neonatal complications were predominantly concentrated in LBW and VLBW categories, with a rising gradient as birthweight decreased. The main complications included respiratory distress and NICU admissions.

## Figures and Tables

**Figure 1 medicina-60-00193-f001:**
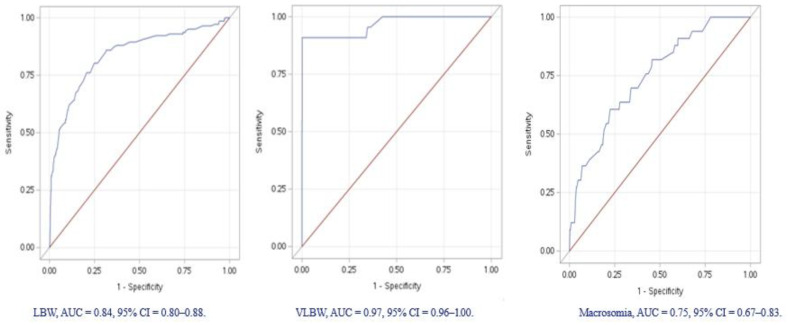
Receiver Operating Characteristic curves of birthweight categories using logistic regression.

**Table 1 medicina-60-00193-t001:** Association between birthweight categories and maternal factors.

		Birthweight Categories								*p*-Value
		NBW N = 1638		LBW N = 153		VLBW N = 27		Macrosomia N = 37		
		N	%	N	%	N	%	N	%	
**Mother Age**	**<30 years**	816	49.94	75	49.34	13	50.00	17	45.95	0.94
	**≥30 years**	818	50.06	77	50.66	13	50.00	20	54.05	
**Parity**	**Primiparous**	902	55.07	84	54.90	20	74.07	21	56.76	0.27
	**Multipara**	736	44.93	69	45.10	7	25.93	16	43.24	
**BMI**	**Normal**	552	35.57	65	44.52	8	33.33	6	18.18	<0.01
	**Overweight**	543	34.99	50	34.25	7	29.17	8	24.24	
	**Obesity**	457	29.45	31	21.23	9	37.50	19	57.58	
**Gestational Age**	**Early preterm**	10	0.62	20	13.33	24	88.89	0	0.00	<0.01
	**Late preterm**	61	3.75	48	32.00	1	3.70	0	0.00	
	**Early term**	432	26.57	53	35.33	1	3.70	8	21.62	
	**Full term**	1107	68.08	28	18.67	1	3.70	28	75.68	
	**Post term**	16	0.98	1	0.67	0	0.00	1	2.70	
**Diabetes**	**No diabetes**	1468	89.62	140	91.50	23	85.19	27	72.97	<0.01
	**GDM/PGDM**	170	10.38	13	8.50	4	14.81	10	27.03	
**Hypertension**	**No**	1594	97.79	135	88.24	21	80.77	34	91.89	<0.01
	**Yes**	36	2.21	18	11.76	5	19.23	3	8.11	

NBW (normal birthweight), LBW (low birthweight), VLBW (very low birthweight). BMI (Body Mass Index), GDM (gestational diabetes), PGDM (pregestational diabetes). All reported percentages are the valid percentages after considering missing data at each variable level.

**Table 2 medicina-60-00193-t002:** Adverse neonatal outcomes by birthweight categories, N = 1855.

Variables	NBW (N = 1638)	LBW (N = 153)	VLBW (N = 27)	Macrosomia (N = 37)	*p* Value
	Freq. (%)	Freq. (%)	Freq. (%)	Freq. (%)	
**NICU Admission**	282 (17.26)	73 (47.71)	22 (81.48)	8 (21.62)	<0.01
**Respiratory distress**	49 (3.00)	36 (23.68)	16 (59.26)	1 (2.70)	<0.01
**Transient tachypnea**	62 (3.79)	14 (9.21)	1 (3.70)	4 (10.81)	<0.01
**Resuscitation**	89 (5.44)	20 (13.16)	12 (44.44)	2 (5.41)	<0.01
**Mechanical ventilation**	113 (6.92)	37 (24.34)	16 (59.26)	4 (10.81)	<0.01
**CPAP/HFNC > 2 h**	45 (2.77)	37 (24.18)	16 (59.26)	0 (0.00)	<0.01
**FIO2 > 30% ≥ 4 h**	15 (0.92)	10 (6.62)	5 (19.23)	1 (2.70)	<0.01
**Received surfactant**	10 (0.61)	6 (3.95)	5 (19.23)	0 (0.00)	<0.01
**Arterial cord acidosis**	8 (0.56)	0 (0.00)	1 (5.88)	0 (0.00)	0.04
**Hyperbilirubinemia**	364 (22.26)	52 (34.21)	10 (37.04)	7 (18.92)	<0.01
**Initiated breast feeding**	1173 (72.32)	92 (61.74)	9 (33.33)	20 (55.56)	<0.01
**Feeding difficulty**	26 (1.59)	4 (2.63)	2 (7.41)	0 (0.00)	<0.01
**Sepsis**	21 (1.28)	2 (1.32)	5 (18.52)	0 (0.00)	<0.01
**Stillbirth**	1 (0.06)	2 (1.33)	4 (17.39)	0 (0.00)	<0.01
**CS**	480 (29.30)	70 (45.76)	16 (59.26)	19 (51.35)	<0.01
**Mean gestational age ***	39 ± 1	36 ± 2	29 ± 4	40 ± 1	<0.01
**Mean Apgar score ***	9 ± 0.7	8.7 ± 1.2	6.2 ± 1.2	8.9 ± 0.28	<0.01

Freq = frequency, NBW = normal birthweight, LBW = low birthweight, VLBW = very low birthweight, CS = cesarean section. * These two variables are expressed as average ± Standard deviation.

**Table 3 medicina-60-00193-t003:** Characteristics of stillbirth and neonatal death.

Number	Gender	Mode of Delivery	Gestational Age	Birthweight	Stillbirth/Neonatal Death	Motherage	Parity	Associated Risk Factor/Cause of Death
1	male	emergency CS	38	2430	Stillbirth	33	2	IUDF abruptio placenta
2	male	vaginal delivery	25	445	Stillbirth	33	0	IUDF Mother had severe PET, HELLP syndrome
2	male	emergency CS	37	3720	Stillbirth	36	0	IUDF Mother had severe PET and admitted to ICU
2	female	vaginal delivery	31	1300	Stillbirth	40	0	IUFD unexplained
4	female	emergency CS	37	2250	Neonatal death	21	0	IUGR. Neonatal sepsis, admitted to NICU for poor feeding, developed apnea then died
5	male	vaginal delivery	33	2690	Stillbirth	25	1	IUDF Mother has thrombophilia abruptio placenta
6	male	emergency CS	28	1200	Neonatal death	35	0	Respiratory complications of prematurity
7	male	vaginal delivery	33	3010	Stillbirth	27	0	IUFD congenital heart disease
8	female	vaginal delivery	23	550	Neonatal death	23	0	Respiratory complications of prematurity
9	female	vaginal delivery	36	2100	Stillbirth	32	2	IUFD unexplained
10	male	vaginal delivery	34	2400	Stillbirth	29	2	IUFD multiple congenital anomalies
11	male	vaginal delivery	35	2040	Stillbirth	35	0	IUFD IUGR abruptio placenta
12	male	vaginal delivery	32	770	Neonatal death	25	0	Diaphragmatic hernia, hypoplasia of the lung, sepsis, and acidosis
13	female	emergency CS	23	540	Neonatal death	24	3	Mother PET with HELLP syndrome. Respiratory complications of prematurity

IUFD = Intrauterine fetal death, IUGR = Intrauterine growth restriction, CS = Caesarean section, HELLP syndrome = Hemolysis, Elevated Liver enzymes and Low Platelets.

**Table 4 medicina-60-00193-t004:** Maternal factors associated with low birthweights and macrosomia using logistic regression analyses.

Variables	LBW		VLBW		Macrosomia *	
	OR	95% CI	OR	95% CI	OR	95% CI
**Multipara**	0.94	0.59–1.50	0.14	0.01–1.72	0.89	0.40–1.96
**Overweight 28.5–32.9 kg/m^2^**	0.76	0.48–1.19	0.89	0.12–1.76	1.23	0.42–3.61
**Obesity ≥ 33 kg/m^2^**	0.54	0.32–0.91 *	0.59	0.07–5.16	3.18	1.24–8.14 *
**Mother age ≥ 30 years**	0.84	0.52–1.35	0.42	0.04–4.03	1.21	0.54–2.73
**Gestational hypertension**	3.50	1.60–7.63 *	5.83	0.69–49.36	-	-
**Diabetes (GDM/PGDM)**	0.52	0.25–1.08	5.65	0.42–75.61	3.75	1.67–8.41 *
**Gestational age**	0.51	0.46–0.57 *	0.41	0.31–0.53 *	1.69	1.24–2.30 *

LBW (low birthweight), VLBW (very low birthweight). Normal birthweight (NBW) is the reference category in the three logistic models. * Hypertension was not included in the macrosomia model.

## Data Availability

Data from this study is available to researchers upon request and approval of Institutional Review Board at Princess Nourah bint Abdulrahaman University (irb@pnu.edu.sa). The request and approval of data sharing are independent from the research team.

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
