# Peer review of "Neonatal Birthweight Spectrum: Maternal Risk Factors and Pregnancy Outcomes in Saudi Arabia"

_medicina, 2024, doi:10.3390/medicina60020193_

Round 1

Reviewer 1 Report

Comments and Suggestions for Authors

Wahabi et al analyzed the maternal risk factors for low and high birthweight in a large population in Saudi Arabia. 

The study is well described, results are interesting. I have only few observations: 

Please explain the abbreviation in the text

Methods: please differentiate materials (study population) and methods. 

"We included.... for the study" should be included in the results

Author Response

We would like to thank our reviewer for his/her valuable comments, here is our point-to-point reply:

Reviewer (1):

The study is well described, results are interesting. I have only few observations:

  • Please explain the abbreviation in the text

Reply: All revised and corrected, highlighted in the text

  • Methods: please differentiate materials (study population) and methods.

Reply: Done

  • "We included.... for the study" should be included in the results

Reply: Added

Reviewer 2 Report

Comments and Suggestions for Authors

Comments on the manuscript

Materials and Methods

-       The authors should describe inclusion and exclusion criteria clearly

-       Did the authors include twin or multiple pregnancies? The same applies to singleton pregnancies with a history of a preterm delivery. These cases for example are a selection bias in the study.

Statistical analysis

-       Please correct Fischer’s exact test.

Results

-       The authors should cross-check the results in table 1. For instance, in NBW group the sum of under 30 / over 30 is 1634 instead of 1638. The same applies for BMI etc.

Discussion

-       The authors should clearly state the main outcome of their study and elaborate on that. In this case maternal hypertension and its association with LBW.

-       Strengths and limitations are clearly stated.

Comments on the Quality of English Language

Minor editing of English language required

Author Response

We would like to thank our reviewer for his/her comments, here is our point-to-point reply:

Materials and Methods

  • The authors should describe inclusion and exclusion criteria clearly

Reply: This paragraph was expanded in the methods section “We included all single pregnancies with gestational age ≥23 weeks based on last menstrual period and/or ultrasound scanning. Twin pregnancies were excluded from this study and cases with missing birth weight measurements were also excluded”.

  • Did the authors include twin or multiple pregnancies? The same applies to singleton pregnancies with a history of preterm delivery. These cases for example are a selection bias in the study.

Reply: We excluded multiple pregnancies and we included those only whose gestational age is 23 weeks or more. In the multivariate analysis we adjusted all the differences according to the gestation age of pregnancy to avoid any reported bias because of the preterm delivery, as statistical adjustment is much better approach to avoid exclusion of large proportion of the cohort and missing the important presentation of this cohort.

Statistical analysis

  • Please correct Fischer’s exact test.

Reply: Corrected

Results

  • The authors should cross-check the results in table 1. For instance, in NBW group the sum of under 30 / over 30 is 1634 instead of 1638. The same applies for BMI etc.

Reply: We reported all variables as extracted from the medical records, we did not use any imputation techniques for the missing data at any level, and all reported percentages are the valid percentages after considering the missing data at each variable level. We added a note to the first table to highlight this point.

Discussion

  • The authors should clearly state the main outcome of their study and elaborate on that. In this case maternal hypertension and its association with LBW.

Reply:

Within the discussion, our emphasis centered on the primary objectives of the study. We initiated this by delving into the findings related to the prevalence of various birthweight categories. Subsequently, we explored the associations with key maternal risk factors, such as diabetes and hypertension. Lastly, we scrutinized diverse outcomes across all birthweight categories. A comparative analysis was conducted, aligning our results with those of both local and international studies. Our aim was to identify discrepancies, trace underlying reasons, and underscore areas warranting further research and clinical attention. We extend a warm welcome for any additional suggestions or points requiring clarification and discussion.

Reply:

  • Strengths and limitations are clearly stated

Comments on the Quality of English Language

Minor editing of English language required

Reviewer 3 Report

Comments and Suggestions for Authors

The study provides a plethora of results and conclusions, but lacks the element of novelty. The associations that were found are already known to the scientific community and most of these details are presented in the introduction. We find that there is little to no scientific benefit from these conclusions.

Author Response

We would like to thank our reviewer for his/her comments, here is our reply:

Comments and Suggestions for Authors

  • The study provides a plethora of results and conclusions, but lacks the element of novelty. The associations that were found are already known to the scientific community and most of these details are presented in the introduction. We find that there is little to no scientific benefit from these conclusions.

Reply:

Despite the existence of studies with comparable findings to ours, we emphasize the crucial significance of our research for both the international and local scientific communities. Our study presents the outcomes of one of the most extensive cohorts in Saudi Arabia, offering profound insights into neonatal outcomes. This information is paramount for shaping policies and facilitating planning for local health authorities. Moreover, in an age dominated by meta-analysis and evidence-building, the substantial size of our cohort contributes to the generation of high-level evidence that accurately reflects variabilities across diverse populations.